# Clinicopathological Features of Non-Small Cell Lung Carcinoma with BRAF Mutation

Andrea Ambrosini-Spaltro [1,*], Claudia Rengucci [2], Laura Capelli [2], Elisa Chiadini [2], Daniele Calistri [2], Chiara Bennati [3], Paola Cravero [4], Francesco Limarzi [1], Sofia Nosseir [5], Riccardo Panzacchi [6], Mirca Valli [7], Paola Ulivi [2] and Giulio Rossi [8]

[1] Pathology Unit, Morgani-Pierantoni Hospital, AUSL Romagna, 47121 Forlì, Italy; francesco.limarzi@auslromagna.it

[2] Biosciences Laboratory, IRCCS Istituto Romagnolo per lo Studio dei Tumori (IRST) "Dino Amadori", 47014 Meldola, Italy; claudia.rengucci@irst.emr.it (C.R.); laura.capelli@irst.emr.it (L.C.); elisa.chiadini@irst.emr.it (E.C.); daniele.calistri@irst.emr.it (D.C.); paola.ulivi@irst.emr.it (P.U.)

[3] Oncology Unit, Santa Maria Delle Croci Hospital, AUSL Romagna, 48121 Ravenna, Italy; chiara.bennati@auslromagna.it

[4] Department of Medical Oncology, IRCCS Istituto Romagnolo per lo Studio dei Tumori (IRST) "Dino Amadori", 47014 Meldola, Italy; paola.cravero@irst.emr.it

[5] Pathology Unit, Santa Maria Delle Croci Hospital, AUSL Romagna, 48121 Ravenna, Italy; sofia.nosseir@auslromagna.it

[6] Pathology Unit, Bufalini Hospital, AUSL Romagna, 47521 Cesena, Italy; riccardo.panzacchi@auslromagna.it

[7] Pathology Unit, Infermi Hospital, AUSL Romagna, 47923 Rimini, Italy; mirca.valli@auslromagna.it

[8] Pathology Unit, Department of Oncology, Fondazione Poliambulanza, 25124 Brescia, Italy; giurossi68@gmail.com

* Correspondence: andrea.ambrosinispaltro@auslromagna.it; Tel.: +39-0543-731703

**Abstract:** (1) Background: BRAF mutations affect 4–5% of lung adenocarcinomas. This study aimed to analyze the clinicopathological features of lung carcinomas with BRAF mutations, focusing on V600E vs. non-V600E and the presence of co-mutations. (2) Methods: All BRAF-mutated lung carcinomas were retrieved from a molecular diagnostic unit (the reference unit for four different hospitals). The samples were analyzed using next-generation sequencing. Statistical analyses included log-rank tests for overall survival (OS) and progression-free survival (PFS). (3) Results: In total, 60 BRAF-mutated lung carcinomas were retrieved: 24 (40.0%) with V600E and 36 (60.0%) with non-V600E mutations, and 21 (35.0%) with other co-mutations and 39 (65.0%) with only BRAF mutations. Survival data were available for 54/60 (90.0%) cases. Targeted therapy was documented in 11 cases. Patients with V600E mutations exhibited a better prognosis than patients with non-V600E mutations ($p = 0.008$ for OS, $p = 0.018$ for PFS); this was confirmed in PFS ($p = 0.036$) when considering only patients who received no targeted therapy. Patients with co-mutations displayed no prognostic difference compared to patients carrying only BRAF mutations ($p = 0.590$ for OS, $p = 0.938$ for PFS). (4) Conclusions: BRAF-mutated lung carcinomas with V600E (40.0%) had a better prognosis than those without V600E. Concomitant co-mutations (35.0%) did not affect the prognosis.

**Keywords:** NSCLC; lung; BRAF; V600E; co-mutation; prognosis; survival

## 1. Introduction

Lung carcinoma is one of the most common tumors globally, representing the second most prevalent malignancy and the leading cause of cancer-related mortality worldwide [1]. Although lung carcinoma is often diagnosed at advanced and inoperable stages, many targeted therapies have been developed over the last few decades with dramatic responses and improvements in prognosis. In non-small cell lung carcinoma (NSCLC), targeted therapies are mainly directed toward specific altered genes that have been thoroughly analyzed, including EGFR, ALK, ROS1, and RET [2]. Targeted therapies against BRAF

(vemurafenib, dabrafenib, and encorafenib) are currently available and widely used in advanced melanomas [3] and recurrent or metastatic thyroid carcinomas [4]. Promising results with BRAF inhibitors have also been reported for colorectal carcinomas [5,6]. In lung carcinomas, BRAF mutations are rare and have not been extensively investigated in the literature [7], with limited data on therapeutic options [8].

BRAF belongs to the rapidly accelerated fibrosarcoma (RAF) group of serine/threonine kinases and plays a crucial role in cell proliferation and differentiation through the mitogen-activated protein kinase (MAPK) signaling pathway [9,10]. When mutated, BRAF may act as an oncogene, leading to the development of cancer [11]. BRAF mutations affect from 2.2% [12] to 4.9% [13] of lung adenocarcinomas. They are usually detected in non-smokers, females [14], and adenocarcinomas with micropapillary architecture [13]. BRAF mutations in the lungs can be subdivided into two major groups: V600E and non-V600E. V600E mutations are located at codon 600, with the conversion of valine to glutamic acid (V600E); less common substitutions at codon 600 include V600K/D/R/M [15]. In contrast to melanoma, where V600E represents the majority of BRAF mutations, in NSCLC, V600E mutations represent only approximately half of the cases, whereas the other half is composed of other mutations, collectively named non-V600E [16]. V600E and V600D/K/R/M may be collectively classified as class 1 mutations, whereas non-V600E mutations are composed of different mutations and may be further classified into classes 2 and 3 [17]. V600E mutations are particularly interesting because they can be targeted using specific drugs, mainly vemurafenib, dabrafenib, and trametinib [18]. Vemurafenib is especially effective in V600 mutations, but not in other BRAF (non-V600) mutations [19]. Dabrafenib and trametinib have been approved as first-line treatments for patients with metastatic NSCLC harboring the V600E mutation [20]. Non-V600 mutations can be treated with specific targeted therapies [21], but patients usually undergo immunotherapy and chemotherapy [22]. However, treatment of non-V600E mutations has not been well documented, with limited clinical data [22].

Studies have reported a discordant prognostic behavior between V600E and non-V600E BRAF-mutated cases. In cases with the V600E mutation, some studies have reported a worse prognosis [13], whereas others have reported a better prognosis [23] or no difference in prognosis [24]. Furthermore, BRAF mutations may be present either alone or in combination with other co-mutations. The co-occurrence of BRAF mutations and co-mutations has been marginally analyzed [23,25], with limited data on its impact on prognosis.

The purpose of our study was to collect a series of lung carcinomas with BRAF mutations and examine their clinicopathological features, with special emphasis on the following: (1) distinction between V600E and non-V600E mutations, (2) distinction between BRAF mutations alone and BRAF with co-mutations.

## 2. Materials and Methods

We collected all advanced/metastatic NSCLCs with BRAF mutations resulting from routine diagnostic molecular analysis performed between 2019 and mid-2022 in AUSL Romagna using next-generation sequencing (NGS). The cases were provided by 4 different hospitals (Cesena, Forlì, Ravenna, and Rimini, Italy). NGS analysis was performed at the Molecular Diagnostics Laboratory of IRST-IRCCS (Meldola, Italy). A dedicated pathologist assessed the neoplastic content (at least 500 tumor cells) and tumor/normal tissue ratio (more than 30%). The pathological features and clinical follow-up data were retrieved.

### 2.1. Clinical Data

The clinical and survival data were retrieved from an electronic database. Progression was evaluated according to the Response Evaluation Criteria in Solid Tumors (RECIST) criteria, version 1.1 [26].

### 2.2. NGS Analysis

We used an amplicon-based DNA/RNA NGS panel, OncomineTM Focus Assay (Thermo Fisher Scientific, Waltham, MA, USA), which can detect 52 genes. The DNA panel could identify hotspot mutations in 35 genes (including BRAF) and copy number variants in 19 genes (including BRAF). The RNA panel was able to detect fusion drivers in 23 genes (including BRAF). We focused only on BRAF-mutated cases and excluded those with BRAF fusion and/or amplification.

To extract DNA/RNA, we used the MagMAX FFPE DNA/RNA Ultra Kit (Applied Biosystems, Waltham, MA, USA) following the manufacturer's protocol, with formalin-fixed and paraffin-embedded (FFPE) material (6–8 tumor sections of 5 μm) or with cytological smears. The DNA concentration was calculated through fluorometric quantitation using a Qubit 4.0 Fluorometer with a Qubit DNA HS (High Sensitivity) Assay Kit (Thermo Fisher Scientific).

DNA library preparation was performed automatically using the library preparer "Ion Chef™ System" (Thermo Fisher Scientific) following the manufacturer's instructions, with 10 ng of input DNA per sample.

The template was then prepared on the Ion Chef™ System (i.e., DNA and RNA from the same sample were combined on the same chip), and sequencing was performed on the Ion S5 Plus platform (Thermo Fisher Scientific) using Ion 520 Chips (Thermo Fisher Scientific).

The primary evaluation was performed using Torrent Suite Software™ (5.12.3) for initial quality control, including chip loading density, median read length, and number of mapped reads. Subsequently, each sample was analyzed using Ion Reporter™ Software (5.16), a suite of bioinformatic tools for variants, filtering, and annotations.

### 2.3. IHC

We used an automated immunostainer (ULTRA, Ventana Medical Systems, Roche, Tucson, AZ, USA) with the following antibodies: VENTANA anti-ALK (D5F3) rabbit monoclonal primary antibody (Ventana Medical Systems, AZ, USA), ROS1 (D4D6) rabbit monoclonal antibody (Cell Signaling Technology, Inc., Danvers, MA, USA), and PD-L1 Dako 22C3 anti-PD-L1 primary antibody (Agilent, Santa Clara, CA, USA). For PD-L1, we relied on a laboratory-based test, which provided results comparable to those reported in the literature, as previously described [27]. Sections of 4 μm thickness were mounted on positively charged slides. EZ Prep solution (Ventana Medical Systems) was used to remove paraffin and reaction buffer to rinse the slides between the staining steps. Antigen retrieval was performed using Cell Conditioning 1 (CC1) (pH 8.0) antigen retrieval solution (Ventana Medical Systems) for 64 min at 95 °C. For ALK, specimens were incubated with primary anti-ALK antibody (prediluted) for 16 min, using OptiView DAB Detection and Amplification. For ROS1, specimens were incubated with primary anti-ROS1 antibody at a concentration of 1:100 for 32 min. For PD-L1, specimens were incubated with primary anti-PD-L1 antibody at a concentration of 1:25 for 64 min at 37 °C, followed by using an OptiView DAB IHC Detection Kit. The slides were stained with hematoxylin and covered with coverslips. Appropriate positive controls were used for each run. In PD-L1 staining, an internal control (normal tonsil) was added to each slide.

The staining was evaluated using a specific scoring system. For ALK, cases were scored as positive or negative, according to the manufacturer's instructions [28]. For ROS1, cases were scored as moderate (2+) or strong (3+) intensity of cytoplasmic staining with at least 50% of neoplastic cells [29]. For PD-L1, cases were considered adequate if at least 100 tumor cells were available. The slides were viewed by one qualified pathologist, with a collegial discussion of the difficult cases. According to the Tumor Proportion Score (TPS), cases were scored for PD-L1 in three groups: (1) less than 1% positive cells (TPS < 1%, negative), (2) 1–49% positive cells (TPS: 1–49%, low expression), and (3) at least 50% positive cells (TPS $\geq$ 50%, high expression) [30].

*2.4. Statistical Analysis*

Overall survival (OS) and progression-free survival (PFS) were estimated using Kaplan–Meier survival curves. OS was defined as the time from the date of molecular analysis (corresponding to the diagnosis of metastatic/advanced disease) to the date of death or the last follow-up for living patients. PFS was defined as the time from the date of molecular analysis to the date of radiological/clinical progression or death from any cause, whichever occurred first, or the last follow-up visit for patients who were alive without disease progression. In the case of no data on disease progression, we used the date of death to calculate PFS. The log-rank test was used to compare OS and PFS between the different subgroups. Median OS and PFS with a 95% confidence interval (CI) were calculated. Multivariate Cox regression analysis was subsequently performed to further evaluate the validity of the significant variables for both OS and PFS.

We examined the differences for V600E vs. non-V600E, and BRAF alone vs. BRAF with co-mutations. We used the Mann–Whitney test for non-parametric variables to compare the values in different categories. Statistical significance (*p*) was set at 0.05, using a 2-tailed hypothesis. Statistical analyses were performed using Microsoft Excel 2020 (Microsoft Corp., Redmond, WA, USA) and SPSS version 25 (IBM Corp., Armonk, NY, USA).

## 3. Results

*3.1. Case Series*

A total of 60 NSCLCs with BRAF mutation were retrieved. One additional case showed only a BRAF copy number alteration (with no BRAF mutation) and was excluded from this series. No cases with BRAF fusion were found.

All cases were NSCLCs, predominantly adenocarcinomas (55 cases, 91.7%), with 2 (3.3%) NSCLCs not otherwise specified (NOS), 1 (1.7%) squamous cell carcinoma, 1 (1.7%) adenosquamous carcinoma, and 1 (1.7%) carcinoma with sarcomatoid features. Detailed information is summarized in Supplementary Table S1 for pathological features (in all 60 cases) and in Supplementary Table S2 for clinical stage (in 44/60 cases) and therapeutic regimens (in 54/60 cases).

Molecular analysis was performed on bioptic material in 32 cases (53.3%), cytological material (smears or cell blocks) in 19 cases (31.7%), and surgical resection in 9 cases (15.0%).

The 60 BRAF-mutated cases were subdivided into 24/60 (40.0%) V600E and 36/60 (60.0%) non-V600E. Among the non-V600E BRAF mutations, the most frequent mutations were the following: G466A, D594N, G469A, and N581S (8,3%, 6.7%, 6.7%, and 6.7% of all BRAF mutations, respectively). A summary of BRAF mutations is illustrated in Figure 1a,b and specifically described in Table 1.

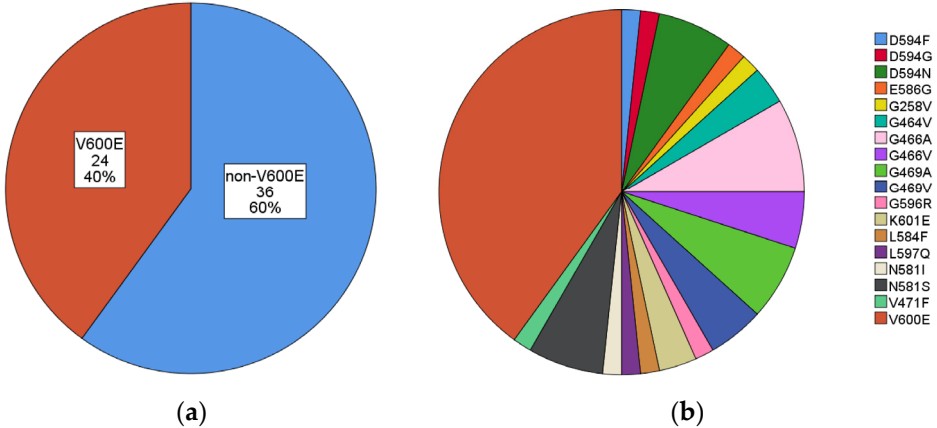

**Figure 1. Summary of identified BRAF mutations.** (**a**) V600E vs. non-V600E; (**b**) V600E and all other non-V600E mutations.

**Table 1.** Summary of all identified BRAF mutations.

| Mutation | N. (%) | Mutation | N. (%) |
|---|---|---|---|
| V600E | 24 (40.0) | V600E | 24 (40.0) |
| | | G466A | 5 (8.3) |
| | | D594N | 4 (6.7) |
| | | G469A | 4 (6.7) |
| | | N581S | 4 (6.7) |
| | | G466V | 3 (5.0) |
| | | G469V | 3 (5.0) |
| | | G464V | 2 (3.3) |
| | | K601E | 2 (3.3) |
| non-V600E | 36 (60.0) | D594F | 1 (1.7) |
| | | D594G | 1 (1.7) |
| | | E586G | 1 (1.7) |
| | | G258V | 1 (1.7) |
| | | G596R | 1 (1.7) |
| | | L584F | 1 (1.7) |
| | | L597Q | 1 (1.7) |
| | | N581I | 1 (1.7) |
| | | V471F | 1 (1.7) |
| Total | | | 60 |

BRAF mutation was present alone in 39/60 (65.0%) cases and in combination with other co-mutations in 21/60 (35.0%) cases. Associated co-mutations included predominantly KRAS, PIK3CA, and EGFR (38.1%, 23.8%, and 9.5%, respectively). Specifically, co-mutations involved the following genes: KRAS (8/21), PIK3CA (5/21, 1 with KRAS as well), EGFR (2/21, 1 with KRAS as well), IDH1 (2/21), FGFR3 (2/21), MET (1/21 with KRAS as well), GNAQ (1/21), CDH4 (1/21), AKT1, and MTOR (1/21). Among V600E-mutated cases, co-mutations were detected in 6/24 cases (25.0%), whereas among non-V600E cases, co-mutations were slightly elevated, representing 15/36 cases (41.7%). The types and frequencies of specific co-mutations are summarized in Table 2.

Using IHC, ALK and ROS1 were available in 42 cases, and both were negative in all 42 cases. PD-L1 was determined in 53 cases, with the following results: 7 (13.2%) negative cases (TPS < 1%), 25 cases (47.2%) with low expression (TPS: 1–49%), and 21 cases (39.6%) with high expression (TPS ≥ 50%).

Follow-up data on mortality and progression were available for 54/60 (90%) cases, ranging from 0 to 49 months (mean: $13.0 \pm 13.7$, median: 8.0): 22 with V600E mutation and 32 with non-V600E mutations; 18 with co-mutations and 36 without co-mutations. Among the 22 V600E cases with follow-up, 11/22 (50.0%) received specific targeted therapy against BRAF (alone or in combination with chemotherapy) (8 in first-line and 3 in second-line), 5/22 (22.7%) received other therapies, 5/22 (22.7%) received no therapy (only best supportive care), and 1/22 (4.5%) had no data on the specific therapy administered. Among non-V600E cases, no targeted therapy was administered. Overall, 36/60 (60.0%) patients died; one 96-year-old patient was considered censored (died of other causes), whereas the other 35/60 (58.3%) patients were considered to have died of the disease. Progression was observed in 5/60 (8.3%) cases. Detailed information on survival data (OS and PFS) is presented in Supplementary Table S3.

**Table 2. Co-mutations associated with BRAF.** Summary of all detected co-mutations associated with BRAF mutation.

| Gene | | N. (%) | N. (%) |
|---|---|---|---|
| KRAS | | 4 (19.0) | |
| KRAS, EGFR | | 1 (4.8) | |
| KRAS, MET | | 1 (4.8) | |
| KRAS, PAM2K1 | | 1 (4.8) | |
| KRAS, PIK3CA | | 1 (4.8) | |
| | KRAS (all) | | 8 (38.1) |
| PIK3CA | | 4 (19.0) | |
| | PIK3CA (all) | | 5 (23.8) |
| IDH1 | | 2 (9.5) | |
| FGFR3 | | 2 (9.5) | |
| EGFR | | 1 (4.8) | |
| | EGFR (all) | | 2 (9.5) |
| ESR1 | | 1 (4.8) | |
| GNAQ | | 1 (4.8) | |
| CDH4 | | 1 (4.8) | |
| AKT1, MTOR | | 1 (4.8) | |
| Total | | 21 | |

### 3.2. Statistical Analysis

Using the log-rank test, survival analyses revealed a significantly better prognosis in 22 V600E patients than in 32 non-V600E patients, in terms of both OS ($p = 0.008$, Figure 2a) and PFS ($p = 0.018$, Figure 2b). The median OS was 23 months in V600E (95% CI 12.8–33.2) and 6 months in non-V600E (95% CI 2.0–10.0). The median PFS was 17 months in V600E (95% CI 6.0–28.0) and 5 months in non-V600E (95% CI 3.0–7.0). The better prognosis of patients with V600E compared to those with non-V600E was further confirmed when the analysis was restricted to patients who did not receive anti-BRAF targeted therapy (12 cases with V6000E and 32 cases with non-V600E), with significance for PFS ($p = 0.036$, Figure 3b) but not for OS ($p = 0.103$, Figure 3a). In this comparison, we excluded one case from the V600E subgroup since we did not have detailed information on specific treatment. We did not exclude any cases from the non-V600E subgroup because they were not treated with anti-BRAF targeted therapy in the institutions enrolled in this study (anti-BRAF targeted therapy was administered only in V600E cases). Multivariate Cox regression analysis of 40/60 cases showed that V600E ($p = 0.008$ for OS, $p = 0.017$ for PFS) and type of therapy ($p = 0.006$ for OS, $p = 0.009$ for PFS) were significant variables, whereas stage reached statistical significance only for OS ($p = 0.049$) and not for PFS ($p = 0.055$). Detailed information on the Cox regression analysis is summarized in Supplementary Table S4.

In contrast, co-mutations did not affect prognosis, neither in terms of OS ($p = 0.590$, Figure 4a) nor PFS ($p = 0.938$, Figure 4b). The median OS was 8 months in 18 cases with co-mutations (95% CI 0–27.6) and 10 months in 36 cases without co-mutations (95% CI 3.7–16.3). The median PFS was 6 months in 18 cases with co-mutations (95% CI 3.2–8.8) and 6 months in 36 cases without co-mutations (95% CI 0–12.8).

We subsequently examined the differences in various parameters between V600E and non-V600E mutations, and between co-mutated and non-co-mutated cases. Using the Mann–Whitney test, age, sex, histotype, material, smoking status, and PD-L1 expression were variably distributed, with no significant difference between V600E and non-V600E cases, or BRAF alone and BRAF with co-mutations (Table 3). Co-mutations were slightly more frequent in non-V600E cases (15/36, 41.7%) than in V600E cases (6/24, 25.0%); however, this difference was not statistically significant. Values close to significant (but still not significant) were found in comparisons of smoking status, both in V600E vs. non-V600E ($p = 0.08$) and in co-mutated vs. non-co-mutated ($p = 0.07$). However, this comparison was not relevant because the vast majority of these patients were current smokers or ex-smokers, and only 2/35 (5.7%) were non-smokers.

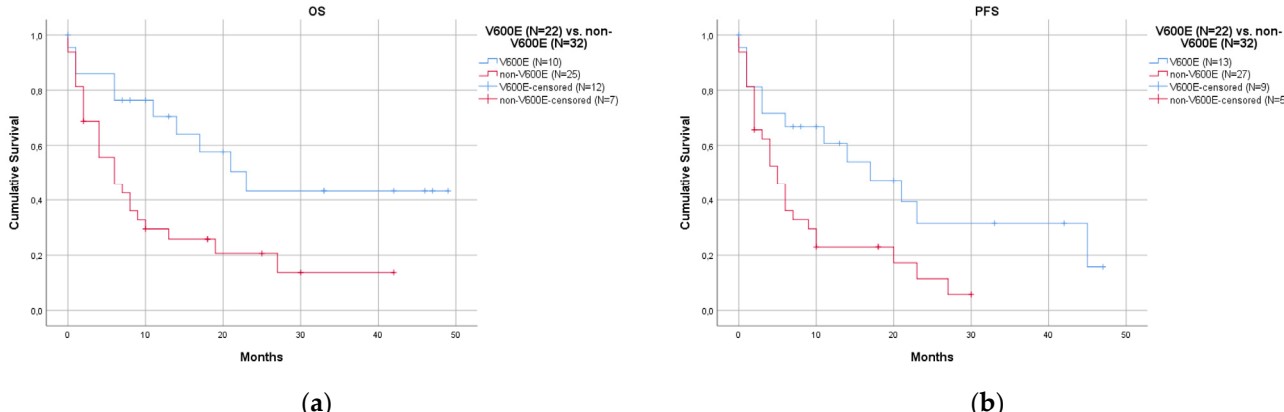

**Figure 2.** Survival curves comparing V600E and non-V600E mutations. Cases with V600E mutations (22 cases, blue lines) showed better prognosis than those with non-V600E mutations (32 cases, red lines): (**a**) overall survival (OS) ($p = 0.008$); (**b**) progression-free survival (PFS) ($p = 0.018$).

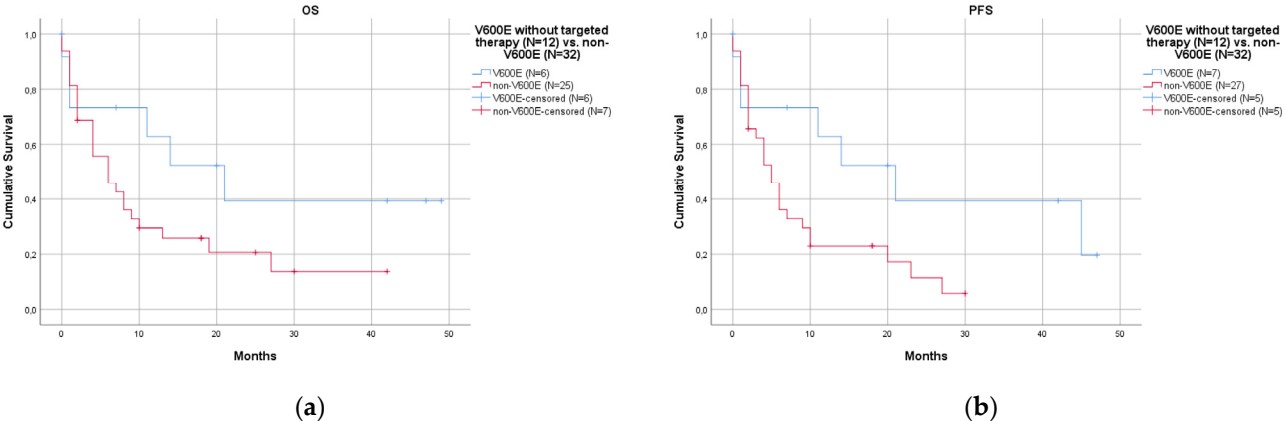

**Figure 3.** Survival curves comparing V600E and non-V600E mutations without targeted therapy. Targeted therapy may be used in BRAF-mutated patients, especially in those harboring V600E. Without targeted therapy, cases with V600E mutations (12 cases, blue lines) exhibited a better prognosis than cases with non-V600E mutations (32 cases, red lines): (**a**) overall survival (OS), without statistical significance ($p = 0.103$); (**b**) progression-free survival (PFS), with statistical significance ($p = 0.036$).

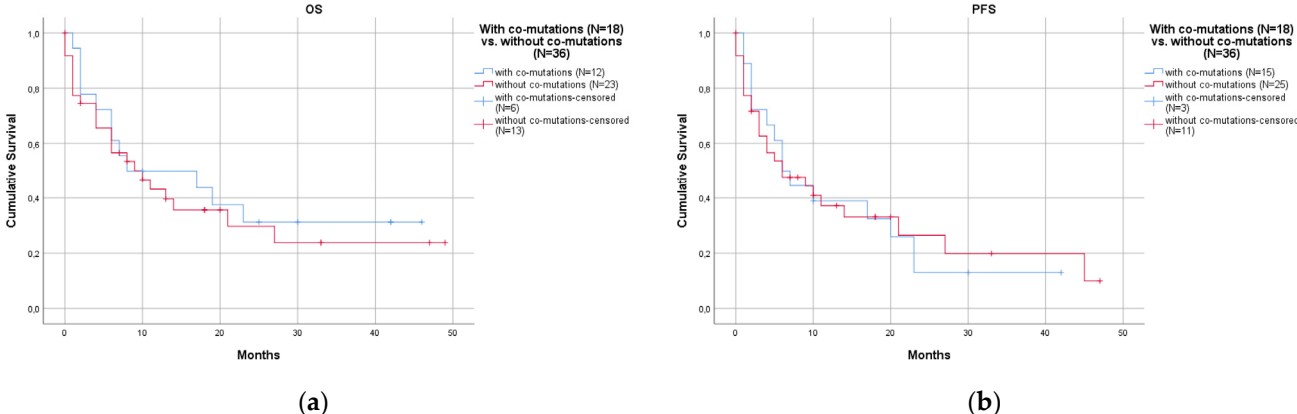

**Figure 4.** Cases with co-mutations (18 cases, blue lines) and without co-mutations (36 cases, red lines) showed no significant differences in prognosis: (**a**) overall survival (OS) ($p = 0.590$); (**b**) progression-free survival (PFS) ($p = 0.938$).

**Table 3.** Clinicopathological features of BRAF-mutated carcinoma, focusing on differences of V600E vs. non-V600E and with vs. without co-mutations. No significant differences were observed in any of the comparisons (SD: standard deviation).

| | | Overall | V600E | Non-V600E | *p*-Value | with Co-Mutations | without Co-Mutations | *p*-Value |
|---|---|---|---|---|---|---|---|---|
| N. | | 60 | 24/60 (40.0%) | 36/60 (60.0%) | | 21/60 (35.0%) | 39/60 (65.0%) | |
| Age (mean ± SD) | | 71.2 ± 9.9 | 71.3 ± 11.1 | 71.1 ± 9.1 | 0.92 | 70.0 ± 10.8 | 71.8 ± 9.5 | 0.74 |
| Sex | male | 37/60 (61.7%) | 12/24 (50.0%) | 25/36 (69.4%) | 0.13 | 15/21 (71.4%) | 22/39 (56.4%) | 0.26 |
| | female | 23/60 (38.3%) | 12/24 (50.0%) | 11/36 (30.6%) | | 6/21 (28.6%) | 17/39 (43.6%) | |
| Histotype | adenocarcinoma | 55/60 (91.7%) | 24/24 (100.0%) | 31/36 (86.1%) | 0.6 | 18/21 (85.7%) | 37/39 (94.9%) | 0.21 |
| | squamous cell carcinoma | 1/60 (1.7%) | 0/24 | 1/36 (2.8%) | | 0/21 | 1/39 (2.6%) | |
| | NSCLC, NOS and other histotypes | 4 (6.7%) | 0/24 | 4/36 (11.1%) | | 3/21 (14.3%) | 1/39 (2.6%) | |
| Material | cytological | 19/60 (31.7%) | 8/24 (33.3%) | 11/36 (30.6%) | 0.76 | 5/21 (23.8%) | 14/39 (35.9%) | 0.97 |
| | bioptic | 32/60 (53.3%) | 11/24 (45.8%) | 21/36 (58.3%) | | 15/21 (71.4%) | 17/39 (43.6%) | |
| | surgical | 9/60 (15.0%) | 5/24 (20.8%) | 4/36 (11.1%) | | 1/21 (4.8%) | 8/39 (20.5%) | |
| Smoking status (data available in 35 cases) | non-smokers | 2/35 (5.7%) | 2/14 (14.3%) | 0/21 | 0.86 | 1/13 (7.7%) | 1/22 (4.5%) | 0.07 |
| | current smokers | 11/35 (31.4%) | 3/14 (21.4%) | 8/21 (38.1%) | | 1/13 (7.7%) | 10/22 (45.5%) | |
| | ex-smokers | 22/35 (62.9%) | 9/14 (64.3%) | 13/21 (61.9%) | | 11/13 (84.6%) | 11/22 (50.0%) | |
| | non-smokers | 2/35 (5.7%) | 2/14 (14.3%) | 0/21 | 0.08 | 1/13 (7.7%) | 1/22 (4.5%) | 0.7 |
| | smokers (current and ex-) | 33/35 (94.3%) | 12/14 (85.7%) | 21/21 (100.0%) | | 12/13 (92.3%) | 21/22 (95.5%) | |
| PD-L1 (total: 53) | TPS < 1% | 7/53 (13.2%) | 3/21 (14.3%) | 4/32 (12.5%) | 0.81 | 4/19 (21.1%) | 3/34 (8.8%) | 0.79 |
| | TPS 1–49% | 25/53 (47.2%) | 9/21 (42.9%) | 16/32 (50.0%) | | 7/19 (36.8%) | 18/34 (52.9%) | |
| | TPS ≥ 50% | 21/53 (39.6%) | 9/21 (42.9%) | 12/32 (37.5%) | | 8/19 (42.1%) | 13/34 (38.2%) | |
| ALK (total: 42) | negative | 42 (100%) | 17 | 25 | | 16 | 26 | |
| | positive | 0 | | | | | | |
| ROS1 (total: 42) | negative | 42 (100%) | 17 | 25 | | 16 | 26 | |
| | positive | 0 | | | | | | |
| Co-mutations | | 21/60 (35.0%) | 6/24 (25.0%) | 15/36 (41.7%) | 0.19 | | | |

## 4. Discussion

This study showed that among BRAF-mutated NSCLCs:

1. Patients with the V600E mutation had a better prognosis than those with non-V600E mutations.
2. Associated co-mutations were not rare (35.0%) and did not affect prognosis.

BRAF mutations are seldom detected in lung tumors, with frequencies varying from 2.2% [12] to 4.9% [13] in lung adenocarcinoma and reaching up to 5.4% [31] in NSCLC. Overall, BRAF mutations in NSCLC show an intermediate frequency between EGFR mutations and ROS1 rearrangements, similar to ALK rearrangement [25], and deserve special attention for the possible use of targeted therapy.

BRAF mutations are subdivided into V600E and non-V600E mutations. In our series (54 cases with FU), we reported a significantly better prognosis for V600E patients than for non-V600E patients, both in terms of OS ($p = 0.008$) and PFS ($p = 0.018$). Studies in the literature have reported conflicting results. In a series of 36 BRAF-mutated cases, Cardarella et al. described no prognostic difference between V600E and non-V600E [24]. Among 36 BRAF-mutated cases, Marchetti et al. reported a worse prognosis in V600E cases [13]. Conversely, in 380 BRAF-mutated cases, Sakai et al. detected a better prognosis in V600E cases [23]. The discrepancies may be due to the evolving role of specific targeted therapies that mainly address V600E cases. In our series, we repeated the survival analyses in V600E patients who did not receive any targeted therapy, and the prognostic advantage of V600E was statistically confirmed only in PFS ($p = 0.036$, Figure 3b); a better prognosis was also evident in OS even if it was not statistically significant (Figure 3a). These conflicting results may also be partially explained by the limited number of cases examined in different studies, given the relatively low frequency of these mutations in the landscape of all NSCLCs. However, the study with the most cases examined in the literature was that described by Sakai [23], who provided results apparently in accordance with ours, underlining the overall better prognosis of cases harboring the V600E mutation.

The better prognosis of V600E cases is clinically important and may be useful in correctly evaluating patients affected by NSCLCs. In NSCLCs, targeted therapy (BRAF/MEK inhibitors) is usually administered to patients carrying the V600E mutation [32]. When evaluating the responsiveness to a specific therapy, the better prognosis intrinsically associated with V600E may represent an important bias. This was also confirmed in our series by the better prognosis of V600E-mutated patients who were not treated with targeted therapy, especially when examined for PFS. For future studies, it may be clinically important to consider the better prognosis associated with V600E cases by not directly comparing them with non-V600E cases or using adequate statistical corrections.

BRAF-mutated NSCLSs may be associated with co-mutations in other genes. In our study, BRAF-mutated NSCLCs were associated with co-mutations in a considerable proportion of cases: 21/60 (35.0%). Co-mutations mainly involved KRAS and PIK3CA, as described by Sakai [23], who also found p53 mutations that were not detected in our series. Sheikine described the co-occurrence of SETD2, SMAD4, and PIK3CA in V600E and of KEAP1, NF1, MET, RICTOR, KRAS, MYC, STK11, and TP53 in non-V600E [25]. The role of co-mutations has recently become more important in the development of cancer, as their presence may change the development of neoplastic disease, its prognosis, and response to therapy [33]. In lung cancer, co-occurring genomic alterations, particularly in TP53 and LKB1 (also known as STK11), have been proven to be core determinants of the molecular and clinical heterogeneity of oncogene-driven subgroups [34]. Co-mutations may modulate responsiveness to immune checkpoint inhibitors [35]. Co-mutations (particularly p53) may reduce responsiveness to anti-EGFR tyrosine kinase inhibitors in EGFR-mutated NSCLCs [36,37]. However, little is known about the prognostic role of co-mutations in BRAF-mutated NSCLCs and their possible role in developing resistance to BRAF inhibitors. The detection of co-mutations may raise suspicion if their presence reduces the efficacy of therapies that specifically target BRAF. In our study, the presence of co-mutations associated

with BRAF did not affect prognosis, neither in OS nor in PFS. Furthermore, the number of co-mutations was slightly higher in non-V600E patients than in V600E patients; however, this difference was not statistically significant. Hence, the presence of co-mutations did not affect the prognosis of BRAF-mutated NSCLCs, without significant differences between V600E and non-V600E cases. A clinical consequence may be that the detection of possible co-mutations may not change the therapeutic approach for BRAF-mutated NSCLCs (both V600E and non-V600E).

In our study, PD-L1 immunoreactivity showed no significant variation among different classes. We did not observe any trend or statistical significance in PD-L1 expression between the V600E and non-V600E subgroups; high PD-L1 expression was present in 42.9% of V600E cases and in 37.5% of non-V600E cases ($p = 0.81$). Perrone et al. found no statistical significance but described a trend towards increased PD-L1 expression in the non-V600E population [31]. They reported high PD-L1 expression in 30% of V600E cases and 47% of non-V600E cases. Dudnik et al. reported a slight association between non-V600E cases and elevated PD-L1 expression (42% in V600E and 50% in non-V600E), even though the values were close to significance ($p = 0.051$) [38]. In contrast, Gibson et al. observed higher PD-L1 expression in V600E (60% of cases) than in non-V600E (39% of cases), but again with no statistical significance ($p = 0.27$) [39]. Independently from PD-L1 expression, recent reports have underlined the important use of immune checkpoint inhibitors in BRAF-mutated NCSLCs, with a limited distinction between V600E and non-V600E [40].

The limitations of this study may be related to the fact that the samples were collected from four different centers using different pre-analytical procedures. However, a centralized molecular laboratory carried out all analyses in this study. Another critical point may be related to the subdivisions of BRAF mutations, which were classified as V600E or non-V600E. A more precise classification subdivides BRAF mutations into three classes [17], with V600 in class 1 and non-V600 in classes 2 and 3. We did not use this more complex classification, even though it is more precise, because we wanted to analyze the impact of V600E on prognosis considering its important role in therapeutic decisions. Finally, the evaluation of PFS may be limited by the fact that when data on progression were lacking, we used the date of death instead of the date of progression. However, most significant analyses were also confirmed for OS, which also plays a more important role than PFS in prognosis evaluation.

## 5. Conclusions

This study shows that BRAF-mutated NSCLCs are a heterogeneous group that can be subdivided into two main subgroups: those with V600E mutations and those with non-V600E mutations. Patients with the V600E mutations show a better prognosis than those with non-V600E mutations. Conversely, the presence of associated co-mutations does not affect prognosis. These features may have clinical implications. Most importantly, the presence of a co-mutation in BRAF-mutated NSCLCs does not seem to affect the natural history of the disease. The conflicting results reported in the literature may be better defined by more extensive studies.

**Supplementary Materials:** The following supporting information can be downloaded at: https://www.mdpi.com/article/10.3390/curroncol30110728/s1, Table S1: Pathological features of all cases; Table S2: Clinical stages and therapeutic regimens of all cases; Table S3: Data on OS and PFS; Table S4: Summary of the Cox regression multivariate analysis.

**Author Contributions:** Conceptualization, G.R. and A.A.-S.; methodology, A.A.-S., C.R. and P.U.; software, A.A.-S. and F.L.; validation, A.A.-S. and P.U.; formal analysis, A.A.-S.; investigation (pathological evaluation), A.A.-S., F.L., S.N., R.P. and M.V.; investigation (molecular analysis), C.R., L.C., E.C. and D.C.; investigation (clinical follow-up), C.B. and P.C.; resources, C.R.; data curation, C.R.; writing—original draft preparation, A.A.-S.; writing—review and editing, P.U. and G.R.; visualization, A.A.-S.; supervision, G.R.; project administration, G.R. and A.A.-S. All authors have read and agreed to the published version of the manuscript.

**Funding:** This research received no external funding.

**Institutional Review Board Statement:** This study was conducted in accordance with the Declaration of Helsinki. Ethical review and approval were waived for this study. Data were already collected for internal controls, completely anonymized, and subsequently organized for statistical purposes. No additional analyses were performed for this specific study. No identified information of patients was used; hence, the ethical approval was not required for this study.

**Informed Consent Statement:** Due to the retrospective nature of this study and since the data were anonymized, the need for informed consent was waived.

**Data Availability Statement:** The data that support the findings of this study are available from the corresponding author upon reasonable request.

**Acknowledgments:** The graphical abstract was partly generated using Servier Medical Art (https://smart.servier.com, accessed on 28 September 2023), provided by Servier, licensed under a Creative Commons Attribution 3.0 unported license.

**Conflicts of Interest:** The authors declare no conflict of interest. The funders had no role in the design of the study; in the collection, analyses, or interpretation of data; in the writing of the manuscript; or in the decision to publish the results.

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
