# Peer review of "Clinicopathological Features of Non-Small Cell Lung Carcinoma with BRAF Mutation"

_curroncol, doi:10.3390/curroncol30110728_

Round 1

Reviewer 1 Report

Comments and Suggestions for Authors

Title: Clinicopathological features of non-small cell lung carcinoma with BRAF mutation.

Summary: Authors examined effect of BRAF V600E mutation presence and absence on NSCLC survival and prognosis. Thay also analyzed the effect of BRAF mutation alone and in combination with other most commonly occurring co-mutations.

Overall study is a good information for better prognosis od NSCLC patients with BRAF mutations.

Study is well done. I have following comments.

Major comments:

1.       Line 168-175: Authors should add a Table for patient demographics which is described in this section of results.

2.       Fig2a, 2b: Please add description of number of patients used in comparison of both groups in both text and Figure.

3.       Fig 3a, 3b: Please add description of number of patients used in comparison of both groups in both text and Figure.

4.       Fig 4 and 5: Also here, please add description of number of patients used in comparison of both groups in both text and Figure.

Reviewer 2 Report

Comments and Suggestions for Authors

This is a retrospective study of patients with BRAF-mutated metastatic NSCLC. The authors stated that the prognosis of V600E is better than that of non-V600E and that concurrent mutations have no effect on prognosis.

The key drugs for advanced NSCLC without actionable mutations are platinum-based cytotoxic chemotherapy and immune checkpoint inhibitors; BRAF mutations are more common in patients with a history of smoking than EGFR or ALK mutations, so in addition to molecular targeted therapy platinum-based cytotoxic chemotherapy and immune checkpoint inhibitors may be important agents for BRAF-mutated patients. Therefore, the authors must report the regimen. We do not know which regimen is responsible for the PFS in this study.

Staging should be reported. Patients with postoperative recurrence generally have a better prognosis than patients with stage IVB disease.

Women and nonsmokers are favorable prognostic factors for cytotoxic chemotherapy and molecular targeted therapy; for immunotherapy, high PD-L1, men, and smoking history are favorable prognostic factors. Again, regimens need to be reported.

Reviewer 3 Report

Comments and Suggestions for Authors

In this manuscript, the authors insisted that patients with the V600E mutation had a better prognosis than those with non-V600E mutations and associated co-mutations were not rare and did not affect prognosis.

This is an interesting study because there are the survival analyses in V600E patients who did not receive any targeted therapy and the prognostic advantage of V600E was confirmed in terms of both OS and PFS.

However, reviewer does not think that concomitant co-mutations do not affect the prognosis because this study is too small. Could you weaken conclusion and so on?

Comments on the Quality of English Language

Minor editing of English language required

Round 2

Reviewer 1 Report

Comments and Suggestions for Authors

Accepted. Congratulations.

Author Response

Thank you very much once again.

Reviewer 2 Report

Comments and Suggestions for Authors

Thank you for presenting detailed data. In table S2, what was the treatment received by the patient whose treatment is not listed? Also, for the treatment listed as "none", is the treatment unknown or did the patient not receive any treatment?

Are stage IA and stage IIB patients relapsed? Did you perform a molecular diagnosis at the time of relapse? Only 11 patients with V600E and 17 patients with non-V600E have treatment details listed for advanced or recurrent NSCLC. The initial treatment regimen is unknown for the 5 patients in the V600E group and 4 patients in the non-V600E group who were diagnosed with surgical specimens. Are these 9 patients definitely recurrent cases?

If the table is correct and all treatments were collected, we cannot discuss overall and progression-free survival with the present data.

Tables S1 and S2 should be integrated. Authors must collect missing data and add PD-L1 expression.

Reviewer 3 Report

Comments and Suggestions for Authors

The paper was nicely revised.

Author Response

Thank you very much once again.

Round 3

Reviewer 2 Report

Comments and Suggestions for Authors

Major comments.

The data of the molecular analysis is very well analyzed and comprehensive. If authors can collect full treatment data, the report will be more informative. If not, the conclusion of survival data especially PFS must be toned down.

I believe that the number of patients in both groups who used IO, received cytotoxic chemotherapy, and those who received treatment, but the detailed regimen is unknown should be described and the large number of unknowns should be added to the limitation of this study in discussion.

 Fig. 2b, 3b, 4b: The difference in PFS gives the impression that it is a difference that might be meaningful, but it is difficult to understand what it means because the regimens are varied and include unknown details. If it is to be listed, it should be a supplement table.

Minor comments.

Table S2: Adjuvant chemotherapy is not a first-line treatment. The following treatment for the three patients who received adjuvant chemotherapy, i.e., first-line treatment, is not listed. Thus, the number of cases with unknown first-line treatment regimens is 21/61, not 18/60; line 174: 39/60, not 42/60; and line 173: 39/60, not 42/60.

Table S2: Cases of postoperative recurrence should be listed as postoperative recurrence in the Stage column of Table S2.

Table S2: The authors mentioned that "none" is no therapy but isn't that synonymous with BSC? there are 11 cases of none, and since the authors mention that the PFS could be evaluated at 54/60(line 204), there are at most 6 cases of no treatment. The authors mentioned "nv" is not evaluable, but either the treatment was given but the detailed regimen is unknown, or it is unknown if the patient was treated or not.

Please review again the description of nv and none.

Line 206: 8 with molecularly targeted therapy, 9 with other therapy, and 5 with missing data. Although it is stated, in Table S2, when the two cases of adjuvant chemotherapy are combined, there are 8 cases with unknown primary treatment regimen and 5 cases with no treatment. There are 3 cases who received only non-molecularly targeted therapy. Are there cases out of nv and none that are known to have not used BRAF inhibitors?

Fig.3: Why is the number of V600E group 11 cases? PFS data was available for 22 patients in the V600E group, and 8 patients received BRAF-targeted agents, so the number should be 12 cases.
